# COVID-19 and Lockdown: The Potential Impact of Residential Indoor Air Quality on the Health of Teleworkers

**DOI:** 10.3390/ijerph19106079

**Published:** 2022-05-17

**Authors:** Ana Ferreira, Nelson Barros

**Affiliations:** 1Department of Environmental Health, Coimbra Health School, Polytechnic of Coimbra, Rua 5 de Outubro—São Martinho do Bispo, Apartado 7006, 3046-854 Coimbra, Portugal; 2UFP Energy, Environment and Health Research Unit (FP-ENAS), Universidade Fernando Pessoa (UFP), Praça Nove de Abril, 349, 4249-004 Porto, Portugal; nelson@ufp.edu.pt

**Keywords:** indoor air quality, health, dwellings, COVID-19, telework

## Abstract

In addition to outdoor atmospheric contamination, indoor exposure to pollutants is a prime contributor to the overall human exposure, and may condition the expressiveness and severity of respiratory, cardiovascular, and allergic diseases. This situation has worsened due to COVID-19, as people have spent more time indoors to comply with social isolation and mandatory telework. The primary purpose of this study was to assess and compare indoor air quality (IAQ) in a significant sample of dwellings of workers from a Higher Education Institution (HEI) in Portugal who were teleworking and their usual workplace. The levels of carbon dioxide, carbon monoxide, and formaldehyde, particles with equivalent diameters of less than 10 μm, 5 μm, 2.5 μm, 1 μm, 0.5 μm, and 0.3 μm, and ultrafine particles, as well as the level of thermal comfort, were measured at both of the sites assessed. It was found that most of the houses studied, as well as the HEI, had good IAQ, although there were places where the concentrations of some pollutants were above the legal standards. On the other hand, a link was identified between the IAQ and the symptoms and diseases observed in the workers who participated in the study. These results offer the opportunity to make corrective interventions, thereby controlling the sources of pollutants and promoting better ventilation in order to reduce the risk for workers.

## 1. Introduction

Our health and well-being are affected by air quality, which can influence our future. Indoor air quality (IAQ) is one of the main environmental risks for public health [1,2]. Indoor air pollution levels are often higher than those of outdoor air, with concentrations that can reach values two to five times, and occasionally up to one hundred times those observed in the building envelope. Indoor air pollution levels become more relevant when it is considered that people spend about 80% to 90% of their time inside buildings [3,4,5]. With telework, due to the COVID-19 pandemic, this tendency to remain in dwellings has increased, and this trend is expected to become the rulein the near future [6,7,8].

Numerous studies report strong associations between pollutant contamination in buildings and health problems, particularly allergies, and, according to the World Health Organization (WHO), many indoor environmental factors, in addition to allergens, such as humidity and air temperature, volatile organic compounds (VOCs) and particulate matter, have significant adverse effects on human health [6,9,10]. Indoor pollution contributes to the overall human exposure, which can be understood as a combination between the local atmospheric exposure and the microenvironments to which each individual is exposed, whether at school, at the workplace, in commercial spaces, or, particularly, in his or her home, since this corresponds to a daily level of exposure that can be extremely significant. In addition to the influence of pollutants from the outside environment, different emission sources inside buildings can be highlighted, such as tobacco consumption, highlighted, fuel burning, materials used in the construction, furniture, decoration, and maintenance of spaces, as well as other products used by its occupants, such as for cleaning [11,12].

Several studies indicate that the most common air pollutants in buildings are particulate matter, formaldehyde (CH_2_O), volatile organic compounds (VOCs), carbon dioxide (CO_2_), carbon monoxide (CO), and airborne bacteria and fungi [13,14,15,16,17,18]. Tahmasebi et al. (2021) showed that with the change in housing occupancy patterns after the COVID-19 outbreak, indoor CO_2_ concentrations may increase significantly [19]. Dominguez-Amarillo et al. (2020) report that at the onset of the COVID-19 pandemic, when compulsory lockdown was enforced in Madrid, Spain, the city’s outdoor air quality improved, but the population’s exposure to indoor pollutants was generally more intense and prolonged. Mainly due to concerns over household energy saving, the lack of adequate ventilation and the more intensive use of cleaning products and disinfectants during the COVID-19 pandemic led to indoor pollutant levels being generally higher than are compatible with healthy environments. The average daily PM_2.5_ concentration increased by approximately 12%, and the average VOC concentration increased by 37% to 559% [20]. Abouleish (2021) points out that worsening indoor air quality may be the result of the home isolation requirement that was in place to reduce the spread of the new coronavirus disease in 2019 (COVID-19), which had a significant impact on human health [21]. Mannan et al. (2021) report that structures, building materials, surface finishes, and general resident activity were the main reasons for the high concentration of VOCs in buildings. Similarly, outdoor particulate concentrations, nearby construction processes, smoking, carpeting, and/or human movement were associated with increased indoor particulate levels [5].

It is known that people spend a considerable proportion of their time indoors, and that this situation worsened with the pandemic crisis in the years 2020 and 2021 with the mandatory lockdown and teleworking. On the other hand, the importance to IAQ in occupants’ health is also known. Thus, the aim of this study was to contribute to the understanding of the impact of indoor air quality in dwellings on the health of workers who were teleworking during the 2020–2021 lockdown when compared to a situation without lockdown, with workers in their usual workplaces.

## 2. Materials and Methods

In this study, measurements of the pollutants legally established in Portuguese legislation were performed. Although there are currently no reference values, given the increasing references to their harmful effects in several studies (including the WHO), particles with equivalent diameters of less than 10 μm (PM_10_), 5 μm (PM_5_), 2.5 μm (PM_2.5_), 1 μm (PM_1_), 0.5 μm (PM_0.5_), and 0.3 μm (PM_0.3_), as well as ultrafine particles, were measured.

The air quality assessments were carried out in homes located in central Portugal, in the room in which the worker performed their telework (kitchens, bedrooms, offices, or living rooms). In total, 4200 measurements were carried out in 70 homes. In addition to these measurements, and when the workers returned to their workplace at the HEI after the mandatory teleworking period, air quality assessments were also carried out in these places. At the workplaces, 1080 measurements were performed at 18 different locations. The number of measurements was lower compared to the number of measurements taken at the workers’ homes, as many of the workers worked in shared, open-space offices. In each dwelling and workplace, information was also collected on the characterization of the buildings.

A questionnaire was also administered to the workers, aiming to collect information regarding health and habitability status, habits, and lifestyles. The questionnaire was divided into four fundamental parts: the first addressed the sociobiographical characterization of workers; the second was intended to obtain data on their health condition, namely, whether they had a history of chronic illness and symptoms of respiratory and allergic disease; in the third part, the habitability conditions were evaluated; the fourth part aimed to obtain information about the workers’ habits and lifestyles.

The questionnaires were carried out after authorization from the President and the Data Protection Office of the IES and were only administered after the consent of the workers was obtained. To comply with ethical standards, the research team was responsible for guaranteeing the confidentiality of the data during its statistical treatment, interpretation, and analysis.

Outdoor air quality measurements were also performed in the areas surrounding the homes (2100 measurements) and workplaces (180 measurements) of the workers who participated in this study. The study protocol was approved by the IES data protection officer.

### 2.1. Measuring Equipment

Portable equipment was used for the measurements, allowing the direct reading of the parameters, air temperature (T), relative humidity (RH), air speed, CO_2_, CO, CH_2_O, PM_10_, PM_5_, PM_2.5_, PM_1_, PM_0.5_, PM_0.3_, and ultrafine particles. The equipment and its main technical characteristics are identified in Table 1.

The thresholds of protection from physical-chemical pollutants were adopted, according to Portuguese legislation (Ordinance No. 138-G/2021, 1 July 2021) [22]. Regarding the thermal comfort parameters, also according to the Portuguese legislation, the Decree-Law nº. 243/86, from 20 August, was used [23].

### 2.2. Assessment Periods

The indoor air quality assessment was carried out in 30-min series, with sampling every minute, both in the morning and in the afternoon. Measurements were performed when the workers were teleworking at home (February to May 2021), and at the workplace, during the months of June and July 2021, during which, due to the non-mandatory nature of teleworking, the workers were already back at their workstations.

Measurements were taken so as to be representative of the air inhaled by workers, away from walls and obstacles, and approximately at the height of the workers’ airways when they were seated.

At this stage, the characteristics of the space were also considered, such as the assessed site area, its volume, the condition of the windows and doors (i.e., whether they were open or closed), the presence of artificial ventilation, heating, and potential sources of pollution/contamination. The outdoor air quality measurements took place near an indoor air intake at the same time as the IAQ measurements.

### 2.3. Statistical Analysis

This was a level II, observational, cross-sectional timeline study. The sampling was of a non-probability type and used the convenience technique. The collected data were treated using the statistical software SPSS, version 27.0.

The interpretation of the statistical tests was based on a significance level of *p* = 0.05, with a 95% reliability margin. As for the descriptive statistical measures, to characterize the analytical evaluations, as well as the structural and physical indicators of the evaluated workers’ spaces, central tendency and absolute dispersion measures were used, as well as frequency analysis (n and %). The statistical tests applied were Pearson’s chi-square and ratio of cross products (odds ratio). The estimates obtained based on the statistical tests were performed at a 95% confidence level for a random error of ≤5%.

## 3. Results

As previously mentioned, the dwellings of 70 HEI workers who were teleworking during the years 2020 and 2021 were evaluated. Most of the workers lived in the city center (41.4%), followed by suburban areas (38.6%). Most of the homes evaluated were between 11 and 30 years old (41 homes). In total, 35.0% of the houses were on the ground floor, and frequently showed direct signs of humidity (91.4%), especially in the rooms where the workers performed their teleworking duties. All the houses had natural ventilation and the windows were double-glazed. In total, 30% had air conditioning (of the fan-convector type) and 51.4% had heating systems, mostly fireplaces designed to burn biomass, usually wood (about 40% had a closed firebox with heat recovery and 60% were open fireplaces). At the time the air quality assessments were performed, there were two or three people in most of the houses (81.5%), and in 92.9% of the spaces, the windows were open. Regarding the weather conditions, on most days, it was sunny with a clear sky (72.9%). The workstations of the workers were also evaluated. The 70 workers carried out their activities in 18 different divisions. Most of the workplaces assessed had two or more workers at the same time when the measurements were taken. The air conditioning systems (of the fan-convector type) were off in 94.4% of the evaluated sites. The workers who participated in the study were non-teaching professionals aged between 36 and 55 (88.6%), and the majority were female (48 workers). Most of them had a college degree (78.6%). Most of the workers did not smoke, and those who reported smoking did not smoke inside the buildings. Outdoor air quality measurements were also taken in the areas surrounding the homes and workplaces.

Four of the environmental parameters assessed (CO_2_, CH_2_O, PM_2.5_, and PM_10_) inside the buildings (housings and workplaces) exceeded the protection threshold established by law, as well as air temperature and relative humidity. Therefore, it was decided to study these parameters in detail in Section 3.1, Section 3.2 and Section 3.3 presents the relation between the indoor air quality in the buildings and the outdoor air quality, considering all the environmental parameters evaluated and not only the parameters that exceeded the values established in the legislation. The results of the evaluation of the workers’ health conditions are presented in Section 3.4. It should also be noted that inside the buildings (houses and workplaces) the average value of the air velocity was <0.001 m·s^−1^, both during the morning and during the afternoon. Regarding the average values outside, they were1.5 m·s^−1^ during the morning and 1.1 m·s^−1^ during the afternoon in the areas around the houses, and 1.7 m·s^−1^ during the morning and 1.3 m·s^−1^ during the afternoon in the areas outside the workplaces.

### 3.1. Concentration of the Various Environmental Parameters Evaluated Inside and Outside the Housing

Figure 1, Figure 2, Figure 3, Figure 4, Figure 5, Figure 6, Figure 7 and Figure 8 show the measurement results of the CO_2_, CH_2_O, PM_2.5_, PM_10_, air temperature, and relative humidity, both inside and outside the houses. In total, 118 measurements of CO_2_ were above the legal protection threshold (corresponding to five houses), and 21 measurements of CH_2_O exceeded the legal protection threshold (corresponding to 17 houses). Regarding PM_2.5_, 981 measurements exceeded the protection threshold (corresponding to 28 houses) and, for PM_10_, 235 measurements exceeded it (corresponding to 21 houses). Regarding the assessments carried out outside the houses, of the 2100 measurements, and regarding the various particles assessed, PM_10_ manifested the highest values.

Bearing in mind the findings inside the dwellings, there was a need to analyze the legal compliance of the physical-chemical pollutants’ measurement results, by means of the following observation:

General criterion [Pollutant]_Max_ ≤ [Pollutant]_LP_, according to which:

[Pollutant]_LP_ corresponds to the protection threshold of the pollutant;

[Pollutant]_Max_ is the maximum value of the average concentrations obtained at all the sampling points.

For both existing and new buildings without mechanical ventilation systems, a tolerance margin (MT) was considered according to the compliance criterion [Pollutant]Max ≤ [Pollutant]LP x (1 + MT), whereby the MT is expressed as a percentage established for each pollutant, according to Table I in Annex I of Ordinance 138-G/1 July 2021.

Thus, for PM_2.5_, 156.2 ≤25.0×(1+1)=156.2 ≤50.0, which does not meet the specific compliance requirement. For PM_10_, 194.2 ≤50.0×(1+1)=194.2 ≤100,0, which does not meet the specific compliance requirement. For CO_2_, 1540.0 ≤1250.0×(1+0.3)=1540.0 ≤1625.0, which meets the specific compliance requirement. As far as the outdoor values are concerned, air temperature reached a maximum value of 30.3 °C and the relative humidity reached a maximum value of 73.6%. Figure 9 and Figure 10 show that, of the 1205 measurements taken of the air temperature that were outside the established range (corresponding to 44 houses), 745 were below 18 °C and 460 were above 22 °C. As for the relative humidity, Figure 11 and Figure 12 show that, of the 1309 measurements that were outside the established range (corresponding to 46 houses), 1047 were below 50% and 262 were above 70%.

For the environmental parameters that exceeded the protection threshold established in the legislation (CO_2_, CH_2_O, PM_2.5_, and PM_10_), we also aimed to determine the areas of their residences in which the workers were at the greatest risk. Table 2 shows the ratio between the parameters that exceeded the protection thresholds and the geographic locations of the workers’ homes.

The study revealed that there were five houses where the CO_2_ levels exceeded the protection threshold and that the differences were statistically significant (*p* = 0.014). Regarding CH_2_O and PM_10_, no significant differences were found. However, most workers who experienced high concentrations lived in the city center. Regarding PM_2.5_ and the different geographical locations of the homes, there were no statistically significant differences. However, it was found that the majority of the workers residing in suburban areas and in the city center were those with the highest concentrations.

### 3.2. Monitoring of Air Quality in the Workplace and Surrounding Environment

As soon as teleworking was suspended, the evaluations in the workplaces and their surroundings were carried out with the workers in their respective workplaces.

Figure 13, Figure 14, Figure 15, Figure 16, Figure 17 and Figure 18 present the results of the various environmental parameters evaluated inside the buildings, namely the average, the median, and the maximum value measured. It was decided to perform the same assessment that was made for the analysis of the results found in the dwellings, using the indoor air quality legislation, considering that the main goal of the study was to safeguard public health, not only of the workers, but of the entire community that stayed in or visited the spaces under assessment.

High maximum values were found among the 1080 measurements taken at the workplaces, as follows: CO_2_, 1348.0 ppm; CH_2_O, 2.4 ppm; PM_2.5_, 171.9 μg·m^−3^; PM_10_, 370.7 μg·m^−3^; air temperature, 29.1 °C; and relative humidity, 77.7%.

Furthermore, it was decided to determine which workers experienced concentrations of pollutants that exceeded the legally established protection threshold. Thus, Table 3 shows the number of measurements for each of the parameters that have a stipulated protection threshold and that, at some point in the evaluations, exceeded this threshold (maximum values), putting the workers at risk.

The analysis of Table 3 shows significant differences between the absence or presence of risk in the different offices and services where the workers performed their activities. In this sense, it was also found that the two workers who experienced high concentrations of CO_2_ performed their duties in the same workspace, although, when applying the tolerance margin, compliance was verified. In terms of PM_2.5_, 11 workers experienced high concentrations (in six workplaces), and there were six workplaces where the legal value of PM_10_ was exceeded, and where 12 workers worked. When the tolerance margin for PM_2.5_ was applied, it was met in only one workplace. As for PM_10_, when applying the tolerance margin, it appears that the margin of tolerance was met in four workplaces. All the workers were faced with high concentrations of CH_2_O in their workplaces, since the protection threshold value was exceeded in all of them.

In six workplaces, the relative humidity was outside the legislated reference range, affecting seventeen workers. Furthermore, it was found that the air temperature in eight workplaces, where twenty-six workers worked, was outside the range defined as adequate (18–22 °C). When the absence and presence of risk was related to whether the air conditioning (fan-convector type), at the time of the measurements, was on or off, the differences were not significant when it came to CO_2_. As for the remaining parameters assessed in the workplaces (except for CH_2_O), statistically significant differences were found when related to the operation of the air conditioning.

From the analysis of the 180 measurements performed on the various environmental parameters outside (the areas surrounding the workplaces), the maximum values obtained for PM_2.5_ (157.3 µg·m^−3^) and PM_10_ (161.8 µg·m^−3^) stand out. The maximum value for air temperature was 39.2 °C, and for relative humidity, it was 74.4%.

### 3.3. Relation between Indoor Air Quality in Buildings and Outdoor Air Quality

Table 4 shows the ratio between the concentrations of the various environmental parameters assessed in the buildings (housing and workplaces) and the concentrations of the same parameters assessed outdoors.

The indoor/outdoor ratio in the housing varied between 0.5 and 10.0, and in the workplaces, between 0.7 and 5.5. In the dwellings, only the values of PM_5_ and T were higher outside than inside. In the workplaces, the PM_0.3_, PM_0.5_, and T values were higher outdoors than indoors.

### 3.4. Evaluation of the Workers’ Health Conditions

Although this is a short-term study, the conditions were assumed to constitute a potential risk to occupants if sustained over time

It was inferred from the study that most workers felt their general state of health was good (45.7%), but that 27.1% considered it satisfactory. The most frequent signs, symptoms, and pathologies reported by the workers were: allergic rhinitis (32.9%), chronic illness and headaches (28.6%), sneezing attacks (27.1%), itching, burning, and irritated eyes (22.9%), and respiratory diseases (21.4%). When comparing men and women, women were more likely to have chronic diseases, sneezing, allergic rhinitis, dry cough, headache, dizziness, and itching, burning, and irritated eyes.

One of the parameters in which the protection threshold was exceeded was CO_2_. However, and although the margin of tolerance was met, it was decided to evaluate the relationship with the health of the workers. It was found that there were no significant differences when with the workers experienced high concentrations of CO_2_ and any signs, symptoms, or diseases that the workers reported themselves as having. Furthermore, most of the workers were not exposed to high levels of CO_2_.

CH_2_O was yet another factor in which the protection threshold was exceeded. Therefore, we tried to understand its relationship with the workers’ health. no significant relationship was found between the fact that the workers were exposed to the risk and their having signs, symptoms, and diseases. However, 12.5% of those with asthma were exposed to risk, as well as 11.1% of those who reported having wheezing and whistling, 26.3% of those with sneezing attacks, 13.0% of those who reported allergic rhinitis, 27.3% of those who reported having a dry cough, 20.0% of those with headaches, and 18.8% of those who reported having itching, burning, and irritated eyes.

In some buildings, the PM_2.5_ exceeded the protection threshold value and, when the margin of tolerance was applied, this value was not met. Accordingly, an attempt was made to determine whether this had a relationship with the workers’ health. It was found that there were significant relationships in workers who had high concentrations of PM_2.5_ and who had asthma (75.0%), chronic bronchitis (85.7%), sneezing attacks (63.2%), dry cough (90.9%), itching, burning, and irritated eyes (81.3%), and breathing problems (75.0%). With regard to dizziness, and in spite of there being no statistically significant differences, it was found that 60.0% had high concentrations of PM_2.5_ and that, with regard to headaches, 50.0% of the workers who complained of them also experienced high concentrations, although the differences were not significant. In terms of allergic rhinitis, there were no statistically significant differences; however, 47.8% of the workers who had this disease were exposed to high concentrations of PM_2.5_ in their workplace.

The PM_10_ also exceeded the protection threshold value in some of the buildings assessed, and when the margin of tolerance was applied, it was found that it was not met. Hence, we sought to evaluate the existence of a correlation between this variable and the workers’ health. No significant differences were observed in the workers who were exposed to high levels of PM_10_ and who had some type of sign, symptom, or disease. Moreover, most of the workers who reported feeling dizzy (60.0%) were exposed to risk, as well as 37.5% of those who reported having asthma, 42.9% of those who reported chronic bronchitis, 33.3% of those who reported wheezing and whistling, 15.8% of those who reported having sneezing attacks, 17.4% of those who had allergic rhinitis, 36.4% of those who had a dry cough, 18.8% of those who reported having itching, burning, and irritated eyes, and 37.5% of those who reported having breathing problems.

## 4. Discussion

This study aimed to understand whether working at home or at the usual workplace would be more harmful to the health of workers in a HEI. To this end, and seizing the opportunity provided by the pandemic situation that forced teleworking, the IAQ was assessed in the workers’ homes and in their various workplaces, as well as in the areas surrounding these buildings. The IAQ of different types of building can have a major impact on human health, given that people spend more than 90% of their time in buildings. The site-specific analysis of buildings can also be beneficial in identifying the main indoor air pollutant at each location, which is essential for long-term solutions. Accordingly, an indicative risk assessment was performed considering the main air pollutants, all the sources of indoor contaminants, and their health-related effects.

Most of the housing units assessed were habitable; however, the maximum values of CO_2_, CH_2_O, PM_2.5_, and PM_10_ sometimes exceeded the legally established protection threshold, revealing the existence of the sources of these pollutants and problems in air renewal.

It was found that in 118 measurements of CO_2_, the maximum values exceeded the legal protection threshold. This indicated that there may be a relationship with the fact that in these five houses, the fireplaces were working, the windows were closed, and there are more than two people in the rooms evaluated. The values of CO_2_ verified outside may have been related to the fact that the houses were near major highways and several industries. There were also 21 measurements of CH_2_O that exceeded the established protection threshold; however, in two of the houses evaluated, the values of CH_2_O were extraordinarily high, which indicated that this situation may have been related to the fact that the houses had new furniture and some of their rooms had been painted shortly before the study. The values of CH_2_O measured outside the houses may have been related to the industries that were in the vicinity of the houses, namely a wood and furniture construction company and a material recycling company. Regarding PM_2.5_, it was found that 981 measurements (28 houses) exceeded the protection threshold. Of these twenty-eight houses, four stood out for the values of PM_2.5_ obtained, which were very high, which indicated a possible relationship with the fact that the houses with natural ventilation were near roads that were under construction during the various evaluations (sanitation works, with ditches opened). Regarding PM_10_, 235 measurements exceeded the legal protection threshold, and the 4 houses identified were the same houses that had already revealed PM_2.5_ values above the legal value. In the other two houses, it was found that the fireplaces were also working, that there were more than two people indoors, and that the houses were close to roads with high levels of traffic. With regard to the concentration of these particles outside, the higher measured values may have been related either to the various works in progress on the roads near the houses, or to the traffic on other roads that were in the vicinity of the houses.

There were significant changes (*p* = 0.014) regarding CO_2_ according to the housing location. In terms of CH_2_O and PM_10_, it was found that there were no significant differences; however, it was found that most of the workers who were faced with high concentrations lived in the city center, where, predictably, the concentrations of these pollutants were higher. As for the relationship between the PM_2.5_ and the different geographic locations of the dwellings, there were no statistically significant differences; however, it was found that most of the workers living in suburban areas and in the city center were found to be at the highest potential risk. Turner et al. (2020) mention that industrial and urban development has caused, worldwide, an increase in the emission of air pollutants, which responsible for adverse effects on public health and the environment. Major industrial activities, traffic, and urban activity are sources of air emissions of various pollutants, which are considered harmful to health, causing prolonged exposure to chemical, biological, and microbiological contaminants [24].

In 2018, the WHO estimated that almost half of the world’s population (about 3 billion people) used polluting technologies and fossil fuels for cooking, heating, and lighting, which are major sources of indoor air pollution. According to Sarafraz, Sadani, and Teimouri (2018), household particulate matter has indoor and outdoor sources [11]. Consequently, indoor air quality is influenced by outdoor air quality [12], and indoor sources, such as heating and cooling systems, cooking facilities that use fossil fuels (oil, gas, kerosene, coal, and wood), smoke, building materials, furniture, carpets, cleaning products, maintenance products, and personal hygiene [25,26,27].

The relationship between the indoor concentrations of various pollutants and the use of heating systems (most of the workers had a working fireplace or air conditioning) was also studied. It was found that all the houses that had CO_2_ values above the legally established protection threshold had a working heating system; however, the differences were not significant. In terms of CH_2_O, there were also no statistically significant differences, although it was found that in the 17 houses in which the CH_2_O values were above the protection threshold, 15 had a heating system and were functioning. For the particles PM_2.5_ and PM_10_, it was found that there were no significant differences, but most of the houses in which the heating system, namely the fireplace, was used also showed particle values above the legal protection threshold.

In terms of thermal comfort, both air temperature and relative humidity, in some of the measurements obtained inside the houses, revealed inadequate values in several rooms, particularly in those in which the workers chose to telework. The verified values may be related to the poor use and regulation of the air conditioning and fireplaces. Thermal comfort plays a significant role in the productivity of the occupants of any indoor environment, and high degrees of thermal discomfort can result in lost productivity [28,29].

With regard to air temperature, it was found that of the 43 houses with inadequate air temperature, 35 had heating systems; however, no significant differences were found. Regarding relative humidity, it was found that there were significant differences between the houses that had inadequate values of relative humidity and the fact that they had heating systems (*p* = 0.043). It was also observed that the 1205 measurements of air temperature that were outside the established range corresponded to 44 dwellings, and that the 1309 measurements of relative humidity that also did not meet the defined range, corresponding to 46 dwellings. Studies state that the installation of heating, ventilation, and air conditioning systems in offices and other buildings is designed to increase comfort by maintaining ideal levels of humidity (usually between 40% and 60%) and temperature; however, the energy crisis of the 1970s introduced the importance of energy saving in buildings, which eventually led to buildings becoming increasingly airtight. There are several sources of indoor air pollution due to energy saving, including the circulation of reduced amounts of fresh air in air conditioning systems, and to the use of synthetic materials and chemicals in buildings for construction and interior decoration purposes. In a study conducted by Almeida et al. (2010) regarding the characterization of indoor air quality in Portugal (HabitAR study), it was found that a considerable number of the houses assessed (47%) had thermal comfort problems. In approximately 30% of the houses assessed, the temperature values were below 20 °C, which represented a significant decrease in thermal comfort [30].

When comparing the results of the air quality in the housing with the concentrations found outside the housings, it became apparent that these results matched the others, i.e., the indoor air quality was worse than the outdoor air quality. These results highlight the importance that indoor air quality should have, particularly since telework is likely to become more intensified in the coming years. In indoor environments, many pollutants are generated by the infiltration of outdoor air through ventilation due to building materials and occupant activities [24].

In this study, after the evaluation of the quality of the residential indoor air and the quality of the outdoor air during the period of social lockdown, in which the workers carried out their functions on a teleworking basis, we proceeded to analyze the indoor air quality in the usual workplaces of the workers, when they returned to work in person. After the 1080 measurements taken at the workplaces, the existence of maximum values of great concern was verified, namely for CO_2_ (1348.0 ppm), CH_2_O (2.4 ppm), PM_2.5_ (171.9 μg·m^−3^), PM_10_ (370.7 μg·m^−3^), air temperature (29.1 °C), and relative humidity (77.7%). It was also found that there were significant differences when the presence and absence of risk was related to the different offices and services where the workers carried out their activities, as well as when relating some of the indoor environmental parameters to the operation of the air conditioning.

It was found that the offices and services where the highest concentrations were analyzed were the computer office, where several people worked simultaneously, and with a significant amount of computer equipment, the office of lending and provisioning, where there were several types of office equipment/material, and the support and secretarial service, where several people worked simultaneously. In these offices, all the windows overlooked a busy highway, with significant emissions of atmospheric pollutants. It should also be noted that most of the facilities where the workers were working had been refurbished shortly before the study (painting, replacement of window frames and light fixtures, etc.), and that their work materials/equipment had been renovated (including desks, cabinets, and computers, among others). These results indicate that the workers’ health and performance may have been at stake. Similar studies concerning administrative environments in recent decades highlight the effects of poor indoor air quality on health, well-being, and work performance [31,32]. There are many sources of indoor pollutants, notably in modern workplaces. These are built with new materials and equipment that emit various chemicals and fine particles, most notably VOCs, namely CH_2_O. Cleaning and hygiene products and flavoring products also contribute to high concentrations of chemical pollutants in the indoor environment. New energy-saving strategies, with the use of heating, cooling, and ventilation systems, also affect the perception of indoor air quality. In addition, the pollutants emitted by office equipment, such as laser printer emissions (ozone, primary VOCs, and particulate matter), and secondary VOCs derived from reactive indoor air chemistry may be of concern [33,34]. In this studythe outdoor air quality was assessed both near residences and workplaces. It was found that, of the 180 measurements of the various pollutants in the outside air, it was the larger particles that manifested the highest values. The air temperature reached maximum values of 39.2 °C and the relative humidity reached maximum values of 74.4%. When comparing the outdoor and indoor environments, it was once again found that the values observed indoors were worse than those outside. Habil et al. (2015) mention that poor indoor air quality can cause various respiratory diseases, allergic diseases, and cancer [35]. Improving IAQ helps to protect human health, reduce absenteeism from work caused by illness, and avoid economic losses caused by medical and hospital treatment.

In addition to these quantitative assessments of the environmental parameters analyzed, both in the houses and in the workplaces (and the surrounding outdoor areas), it was decided to assess the health perceptions of the workers. It was inferred from the survey that most of the workers felt that their general state of health was good (45.7%), but that 27.1% considered it satisfactory. The most frequent signs, symptoms, and pathologies reported by the workers were: allergic rhinitis (32.9%), chronic illness and headaches (28.6%), sneezing attacks (27.1%), itching, burning, and irritated eyes (22.9%), and respiratory diseases (21.4%).

Several studies state that many diseases and death (due to long-term exposure) are related to the inhalation of indoor air pollutants. Most air pollutants contribute to these negative effects on human health due to their particulate nature, especially when the particle diameter is very small, as these particles can lodge in the lung alveoli and quickly pass into the bloodstream [36,37,38]. Ioannis et al. (2021) conducted a study in which they assessed the relationship between the health symptoms reported by building users and indoor pollutant concentrations in a sample of 148 offices as part of the European research project OFFICAIR. The study was conducted in 37 office buildings across eight countries and concluded that the users of offices with higher concentrations of pollutants were more likely to report signs, symptoms, and diseases, including headaches, fatigue, eye irritation, skin problems, and respiratory and cardiac symptoms [39]. In a recent study, in which the authors assessed the correlation between indoor air quality and the health perceptions of workers in an office, a significant correlation was evidenced between respiratory symptoms and increasing indoor particle concentration (especially those of size > 0.3 μm), as well as with several VOCs, namely CH_2_O [40]. The concentration levels of aldehydes, including formaldehyde, were mostly higher indoors (two to five, occasionally a hundred times) when compared to outdoors [41,42], a fact originating from the transformations introduced in the construction of buildings to promote their insulation and airtightness and, consequently, to minimize energy consumption. This situation has even contributed to the Environmental Protection Agency, in the United States, classifying IAQ problems among the main risks to public health and, since 1988, considering formaldehyde as one of the main indoor air pollutants [43]. The results obtained in 2008 by Bartzis, Canna-Michaelidou, and Kotzias (in the BUMA project (prioritization of building materials as indoor pollution sources), in a study developed in the European area, with the purpose of establishing a database concerning the main pollutants emitted by building materials with the potential to influence IAQ, highlighted formaldehyde as one of the pollutants of greatest concern, since the concentrations verified in dwellings were high, often exceeding the established protection thresholds [44]. Since 2006, formaldehyde has been classified as a carcinogen by the International Agency of Research on Cancer [45]. The most obvious health effect is its irritating effect on the eyes and the upper respiratory tract. Formaldehyde concentrations in indoor environments are strongly influenced by the building characteristics, namely the ventilation, the coating, and the finishes used and the decoration, the season of the year (considering that the increase in temperature and humidity encourages formaldehyde volatilization), and the sources of outdoor air [46].

## 5. Conclusions

Given the time spent inside buildings, it is essential to maintain good indoor air quality. In times of lockdown, this issue has become even more important. This study was developed in this context and is intended to be a scientific contribution to a better understanding of the environment–health relationship.

It was found that most of the houses studied were habitable, although the concentrations of various pollutants may suggest that more interventions are required to improve their IAQ by controlling the sources of these pollutants and promoting greater ventilation. It was also found that the air quality inside the buildings, both in the dwellings and in the workplaces, was worse than the outside air quality, and it was also found that the workers experienced values that exceeded the protection thresholds, particularly for the parameters CO_2_, CH_2_O, PM_2.5_, and PM_10_, as well as inadequate thermal comfort values. Some relationships between the IAQ and the signs, symptoms, and illnesses of the buildings’ occupants were also identified.

Although the results obtained are only indicative of workers’ level of risk, it is believed that workers are more exposed to risk inside their homes, either from CO_2_, PM_2.5_, and PM_10_, or from T and RH. However, regarding formaldehyde, the workers were more exposed in the workplace, and this pollutant is one of the most worrying for human health, since it is considered a carcinogen.

In this sense, it is necessary for employers to implement urgent corrective measures. It should be reinforced that the protection thresholds considered for the environmental pollutants under study, defined by Portuguese legislation, are set for non-residential public buildings, so it can be considered that, in residential spaces, it is desirable for occupants to be exposed to even lower values. The workplaces revealed high and worrisome values of CH_2_O, most likely due to recent renovations and new furniture/equipment. Given that CH_2_O is a carcinogenic agent, it is imperative to determine quick measures to eliminate or reduce the concentrations inside the evaluated buildings to levels below the legislated protection threshold value. The composition of the personal hygiene and cleaning products used should be analyzed to determine whether they include compounds that are capable of causing risk to air quality and to the health of building occupants. It is also necessary to examine the possibility of adjusting/limiting the number of employees working in the same spaces on an open-plan office basis. Another measure that can be evaluated is the use of plants, which may have the ability to absorb VOCs and help eliminate the chemicals present in the air, such as chlorophytum, comosum aglaonema, spathiphyllum, dracaena, aloes, or creeper. Regarding the thermal comfort variables, T and RH, control procedures should be adopted so that workers feel comfortable.

It is essential that structural and functional improvements are made in homes and workplaces and that regular monitoring is carried out so as not to expose workers and other building occupants to risky situations. It is also important to improve air renewal systems to make this renewal more efficient and effective. There should be a focus on adequate sun exposure, the avoidance of the appearance of mold and moisture, as well as on the development of awareness, information, and training for all workers so that they adopt healthier, namely by constantly ventilating and airing all the spaces in buildings.

The IAQ should be a priority concern for the government and for all professionals working in public health and occupational health and safety. Considering that telework is likely to become a part of the lives of many professionals, particularly those whose functions allow it, it is necessary to intensify efforts to develop methodologies to improve the conditioning agents of air pollutants that interfere with human health, as well as to create effective tools to improve public health and occupational safety and health, allowing the development of policies in the area of indoor air quality.

The representativeness or significance of the sampled study sites can be generalized beyond the center of Portugal, considering the infrastructural conditions of the buildings in other regions, which are very similar, regardless of their location. However, the generalization of observations should always be made cautiously, since there are variables relevant to IAQ that may vary significantly across the country. Among other factors, the variation in meteorological conditions and associated ventilation practices, as well as the variability in the outdoor air quality across the country, may be relevant.

The results of this study suggest the need for substantial research growth in this field, given the relevant increase in prevalence and severity of diseases that may be related to agents that influence indoor air quality. In this sense, the development of further studies evaluating the impact of air pollutants on the health of the population should be stimulated to contribute to adequate environmental health measures in buildings.

## Figures and Tables

**Figure 1 ijerph-19-06079-f001:**
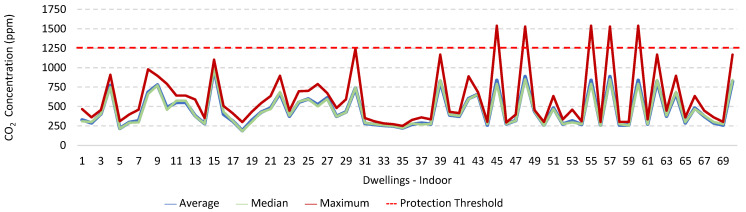
CO_2_ concentration (ppm) indoors.

**Figure 2 ijerph-19-06079-f002:**
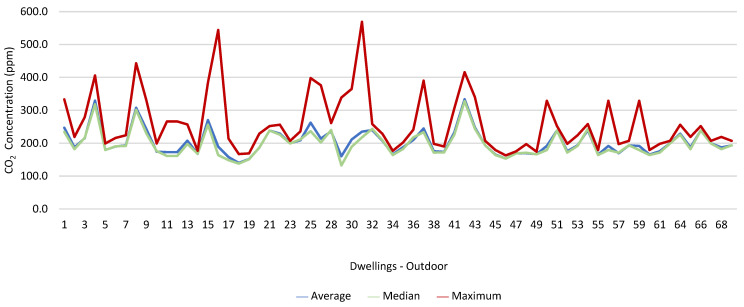
CO_2_ concentration (ppm) outdoors.

**Figure 3 ijerph-19-06079-f003:**
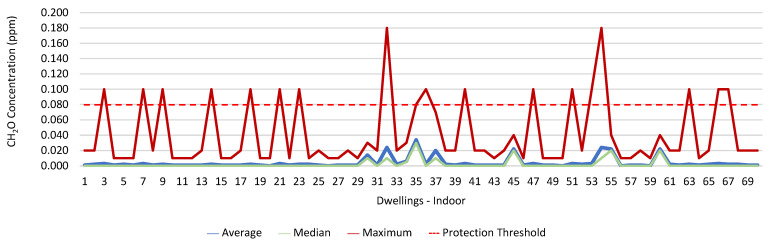
CH_2_O concentration (ppm) indoors.

**Figure 4 ijerph-19-06079-f004:**
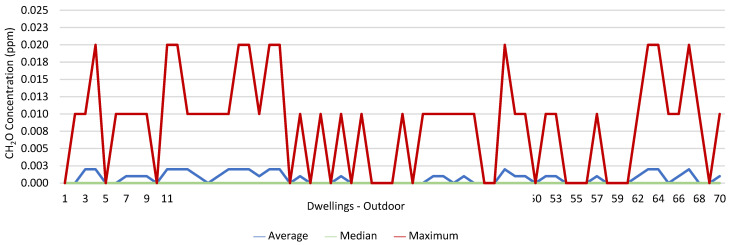
CH_2_O concentration (ppm) outdoors.

**Figure 5 ijerph-19-06079-f005:**
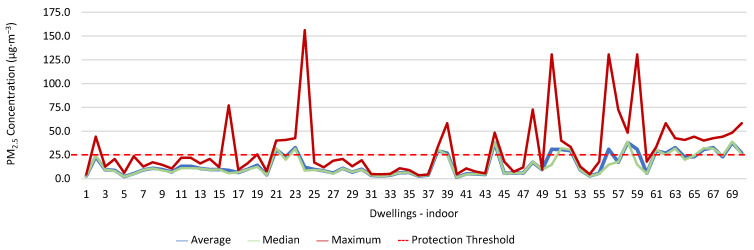
PM_2.5_ concentration (µg·m^−3^) indoors.

**Figure 6 ijerph-19-06079-f006:**
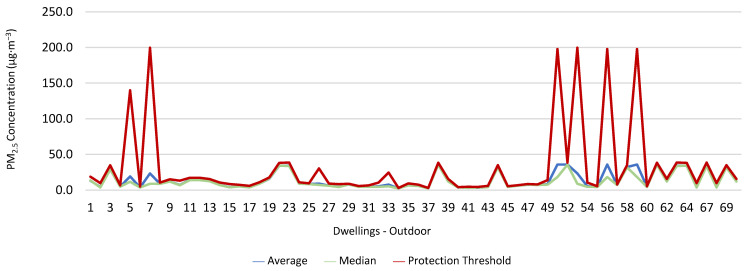
PM_2.5_ concentration (µg·m^−3^) outdoors.

**Figure 7 ijerph-19-06079-f007:**
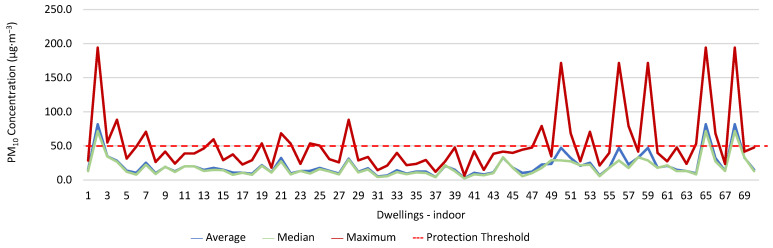
PM_10_ concentration (µg·m^−3^) indoors.

**Figure 8 ijerph-19-06079-f008:**
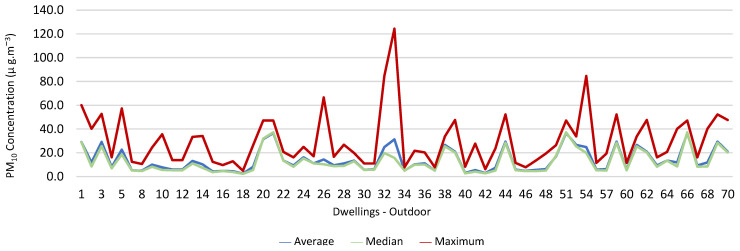
PM_10_ concentration (µg·m^−3^) outdoors.

**Figure 9 ijerph-19-06079-f009:**
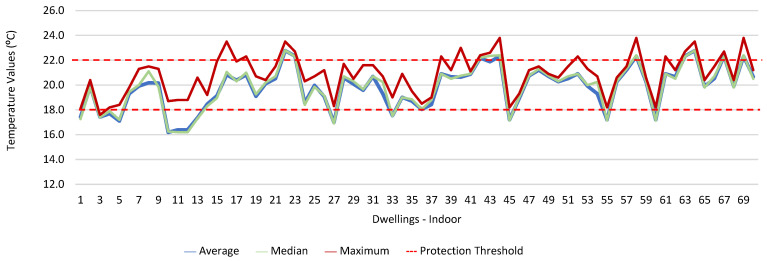
Temperature values (°C) indoors.

**Figure 10 ijerph-19-06079-f010:**
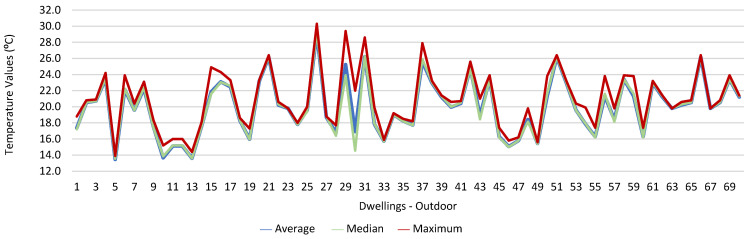
Temperature values (°C) outdoors.

**Figure 11 ijerph-19-06079-f011:**
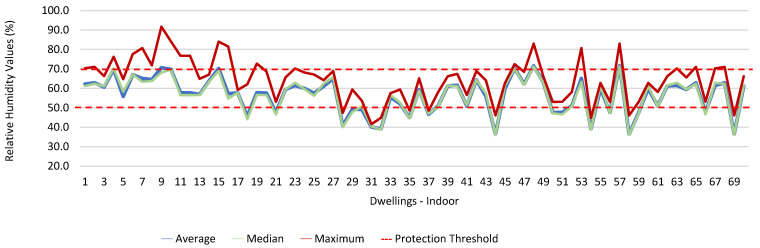
Relative humidity values (%) indoors.

**Figure 12 ijerph-19-06079-f012:**
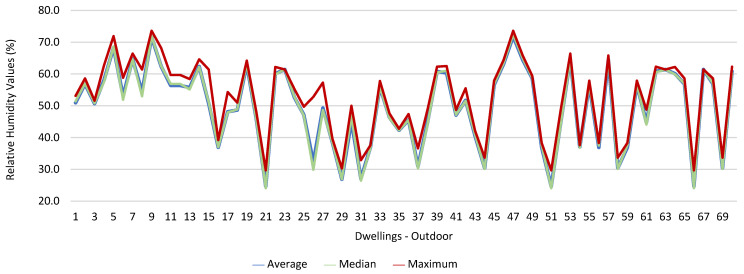
Relative humidity values (%) outdoors.

**Figure 13 ijerph-19-06079-f013:**
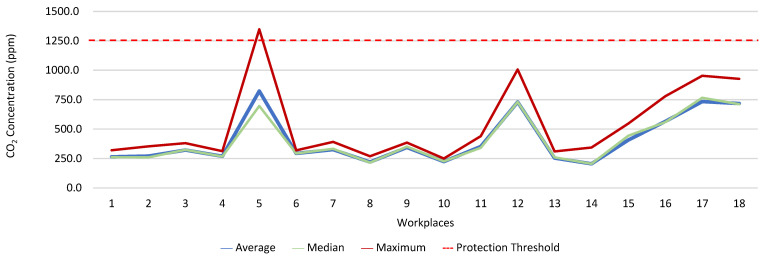
CO_2_ concentration (ppm) inside workplaces.

**Figure 14 ijerph-19-06079-f014:**
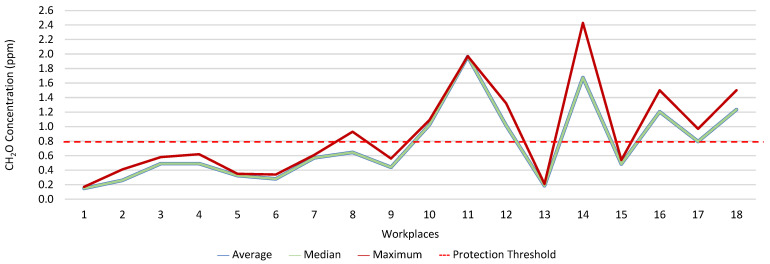
CH_2_O concentration (ppm) inside workplaces.

**Figure 15 ijerph-19-06079-f015:**
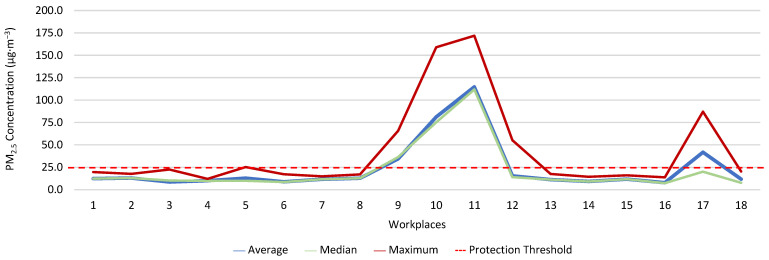
PM_2.5_ concentration (µg·m^−3^) inside workplaces.

**Figure 16 ijerph-19-06079-f016:**
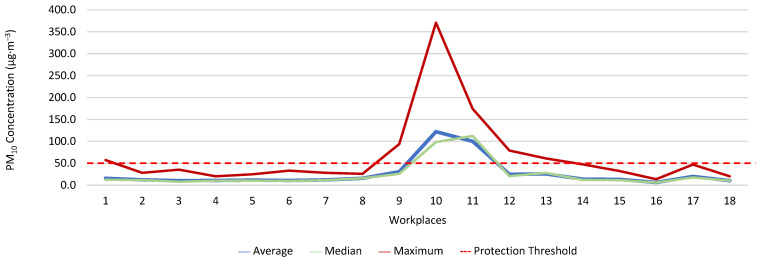
PM_10_ concentration (µg·m^−3^) inside workplaces.

**Figure 17 ijerph-19-06079-f017:**
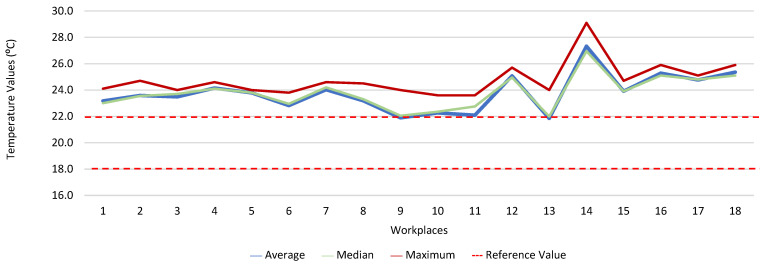
Temperature values (°C) inside workplaces.

**Figure 18 ijerph-19-06079-f018:**
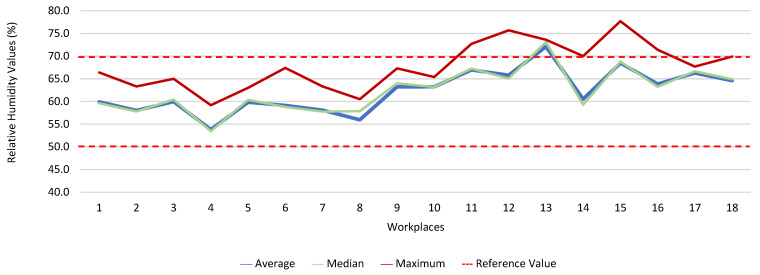
Relative humidity values (%) inside workplaces.

**Table 1 ijerph-19-06079-t001:** Measured parameters, equipment and monitoring methods.

Designation	Equipment	Brand	Locale	Country	Measuring Range	I.M. Monitoring Methods (1)	Method/Reference Principle (2)	Equivalent Methods/Principle (3)
Air velocity evaluation	DELTA OHM HD32.1	DELTA OHM	Caselle di Selvazzano (PD)	Italy	Air velocity sensor AP3203: from 0.05 to 5 m·s^−1^			
Relative humidity evaluation	Q-TrakTM Plus—IAQ Monitor	TSI	Shoreview, MN	U.S.A.	RH: 0 to 95%	Electrochemical Sensor		
Temperature evaluation	Q-TrakTM Plus—IAQ Monitor	TSI	Shoreview, MN	U.S.A.	T °C: 0 to 50 °C	Electrochemical Sensor		
Evaluation of CO_2_	Q-TrakTM Plus—IAQ Monitor	TSI	Shoreview, MN	U.S.A.	CO_2_: 0 to 5000 ppm	NDIR Sensor	Non-dispersive infrared (NDIR) (4)	Electrochemical method; infrared
Evaluation of CO	Q-TrakTM Plus—IAQ Monitor	TSI	Shoreview, MN	U.S.A.	CO: 0 to 500 ppm	Electrochemical Sensor	Electrochemical	Electrochemical method
Evaluation of CH_2_O concentration	PPM FormaldemeterTM htV—IAQ Monitor	PPM Technology	Wales	UK	CH_2_O: 0 to 10 ppm	Electrochemical Sensor	Electrochemical	Electrochemical method
Assessment of airborne particle concentration	Particles Counters	Lighthouse Worlwide Solutions	Kato Road, Fremont	U.S.A.	0 to 8,000,000 particles	Laser sensor	Diode laser	Laser diode method
Evaluation of the concentration of ultrafine particles	P-Track Ultrafine Particle Counter	TSI	Shoreview, MN	U.S.A.	0 to 500,000 particles	Photometric counting of ultrafine particles	Condensation core counter	Condensation core counter sampling method

(1) I.M.—In-house method. (2) Reference method/principle—A reference method is a method established by national, EU or international (e.g., ISO) legislation for the measurement of a specific ambient air pollutant. CEN (EM-ISO) methods are considered reference methods. (3) Equivalent methods/principles. The equivalent method is a measurement method that establishes an adequate response for the purpose in view, with respect to the reference method; in the equivalent method, the results do not differ from the reference method within a certain range of statistical uncertainty. (4) NDIR—nondispersive infrared sensor.

**Table 2 ijerph-19-06079-t002:** Relationship between the parameters that exceeded the protection thresholds and the geographic location of the workers’ homes.

			Building Location			Total
			City Center	Suburban Area	Rural Area	
Risk CO_2_ X^2^; gl; *p* (8.575; 2; 0.014)	Yes	*n*	0	5	0	5
		% Risk	0.0	100.0	0.0	100.0
		% Building location	0.0	18.5	0.0	7.1
	No	*n*	29	22	14	65
		% Risk	44.6	33.8	21.5	100.0
		% Building location	100.0	81.5	100.0	92.9
Risk CH_2_O X^2^; gl; *p* (2.175; 2; 0.337)	Yes	*n*	9	4	4	17
		% Risk	52.9	23.5	23.5	100.0
		% Building location	31.0	14.8	28.6	24.3
	No	*n*	20	23	10	53
		% Risk	37.7	43.4	18.9	100.0
		% Building location	69.0	85.2	71.4	75.7
Risk PM_2.5_ X^2^; gl; *p* (1.561; 2; 0.458)	Yes	*n*	11	13	4	28
		% Risk	39.3	46.4	14.3	100.0
		% Building location	37.9	48.1	28.6	40.0
	No	*n*	18	14	10	42
		% Risk	42.9	33.3	23.8	100.0
		% Building location	62.1	51.9	71.4	60.0
Risk PM_10_ X^2^; gl; *p* (1.895; 2; 0.388)	Yes	*n*	9	6	6	21
		% Risk	42.9	28.6	28.6	100.0
		% Building location	31.0	22.2	42.9	30.0
	No	*n*	20	21	8	49
		% Risk	40.8	42.9	16.3	100.0
		% Building location	69.0	77.8	57.1	70.0
Total		*n*	29	27	14	70
		% Risk	41.4	38.6	20.0	100.0
		% Building location	100.0	100.0	100.0	100.0

X^2^—Pearson test of X^2^; gl—degree of freedom; *p*—*p*-value.

**Table 3 ijerph-19-06079-t003:** Presentation of the measurements in which workers were faced with concentrations that exceeded the protection threshold in the workplace.

Environmental Parameter	Absence|Presence of Risk	Number of Measurements	%	Number of Exposed Workers (Number of Workplaces)	%
CO_2_ X^2^; gl; *p* (1119.776; 17; <0.001)	Absence of risk	1073	99.4	68 (17)	97.1
	Presence of risk	7	0.6	2 (1)	2.9
	Total	1080	100.0	70	100.0
CH_2_O	Absence of risk	0	0	0	0.0
	Presence of risk	1080	100.0	70 (18)	100.0
	Total	1080	100.0	70	100.0
PM_10_ X^2^; gl; *p* (777.196; 17; <0.001)	Absence of risk	963	89.2	58 (12)	82.3
	Presence of risk	117	10.8	12 (6)	17.2
	Total	1080	100.0	70	100.0
PM_2.5_ X^2^; gl; *p* (858.353; 17; <0.001)	Absence of risk	892	82.6	61 (13)	87.1
	Presence of risk	188	17.4	11 (6)	12.9
	Total	1080	100.0	70	100.0
T° X^2^; gl; *p* (384.309; 17; <0.001)	Absence of risk	146	13.5	44 (10)	62.9
	Presence of risk	934	86.5	26 (8)	37.1
	Total	1080	100.0	70	100.0
RH X^2^; gl; *p* (817.114; 34; <0.001)	Absence of risk	987	91.4	53 (12)	75.7
	Presence of risk	93	8.6	17 (6)	24.3
	Total	1080	100.0	70	100.0

X^2^—Pearson test of X^2^; gl—degree of freedom; *p*—*p*-value.

**Table 4 ijerph-19-06079-t004:** Ratio between indoor air quality of buildings and outdoor air quality.

Environmental Parameter	Maximum Values Indoors (Housing)	Maximum Values Outdoors (Housing)	Indoor/Outdoor (Housing)	Maximum Values Indoors (Workplaces)	Maximum Values Outdoors (Workplaces)	Indoor/Outdoor (Workplaces)
CO ppm	2.3	2.1	1.1	2.1	2.1	1
CO_2_ ppm	1540.0	569.0	2.7	1348.0	602.0	2.2
CH_2_O ppm	0.2	0.0	10.0	2.4	0.5	4.8
PM_2.5_ µg·m^−3^	156.2	35.9	4.4	171.9	157.3	1.1
PM_10_ µg·m^−3^	194.2	124.4	1.6	370.7	161.8	2.3
PM_0.3_ µg·m^−3^	24.3	16.7	1.5	7.1	7.6	0.9
PM_0.5_ µg·m^−3^	26.2	9.6	2.7	6.6	6.9	1.0
PM_1_ µg·m^−3^	44.4	21.8	2.0	28.1	26.2	1.1
PM_5_ µg·m^−3^	199.7	412.5	0.5	229.0	105.1	2.2
Ultrafine particles	99.6	34.8	2.9	26.0	4713	5.5
T °C	23.8	30.3	0.8	29.1	39.2	0.7
RH %	91.7	73.6	1.3	77.7	74.4	1.0

## Data Availability

The data presented in this study are available on request from the corresponding author.

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
