# Peer review of "COVID-19 and Lockdown: The Potential Impact of Residential Indoor Air Quality on the Health of Teleworkers"

_ijerph, 2022, doi:10.3390/ijerph19106079_

Round 1

Reviewer 1 Report

 These data obtained with accuracy in Portugal on the effect of smart working on indoor air pollution contains interesting data that although not totally new, expand the general knowledge. The manuscript is rather long, but on the whole it reads well and I only recommend its perusal by a professional writer of English mother language.

Author Response

Thank you very much for your positive comments.

We send a professional writer of English mother language, who made the necessary changes.

Reviewer 2 Report

The manuscript presents a study on the AQ measurements collected for a number of homes and HEI workplaces located in central Portugal. Although the manuscript has some merits, it can be improved in several areas:

(1) The representativeness or significance of the study sites sampled - to what extent the current findings and conclusions may be generalised (i.e., Homes vs. Workplaces, during and after COVID-19 lockdowns) beyond central Portugal?

(2) There is no data/information about the building characteristics of the homes and workplaces where the AQ measurements were taken. For example, (Line 431-432), "where heating system was used, namely fireplace," what type of fireplace was that? Did all the homes use the same type of fireplace?

(3) Exposures and concentrations are different measurements. You seem to use concentrations as a proxy of exposures. If so, some justification is required here.

(4) As the authors' own comment, morbidity and mortality may be attributable to long-term indoor air pollution exposure (line 515). However, the data collected in this study was short term (Feb-May 2021 & Jun-July 2021, Level II, observational, cross-sectional timeline study). Some explanation is required in Section 3.4 as how you link the health observation outcome and the AQ monitoring outcome.

Author Response

Thank you very much for your positive comments.

Reviewer 3 Report

In this paper the authors analyzed the indoor air quality in 70 dwellings of workers from a Higher Education Institution in Portugal who were teleworking during the lock-down period in 2021. In addition, the same analysis was performed at their regular workplace in order to make comparison and level of exposure.  Carbon dioxide, carbon monoxide, formaldehyde, particles of equivalent diameter less than 10 μm, 5 μm, 2.5 μm, 1 μm, 0.5 μm, 0.3 μm and ultrafine particles were measured together with meteorological parameters. Available outdoors concentrations of the related air pollutants were also examined and compared to the indoor values. In order to assess possible impact of the indoor air quality to the specific workers’ health problems a questionary was used to collect information regarding their health, habits, and lifestyles.

Generally, the indoor air quality assessment is of wider interest having in mind recently global trend of teleworking. Although there is no new scientific contribution or novelty of this paper, the obtained experimental results (measurements) still could be useful for future comparison studies.

In the introduction part the goal of the study is clearly defined and most important previous studies and challenges have been analyzed.  The experimental part, the setup and measurement techniques are described with reasonably details. The obtained conclusions are reasonably supported by the results obtained.

There are a few shortcomings that should be addressed prior to publication. Please find below several suggestions that can be used for improving the manuscript.

General comment:

I would recommend to change the title of the manuscript:  based on the descriptive statistics it cannot be assessed the “Impact of domestic indoor air quality on the health of teleworkers”

The causal factors associated with illness cannot be revealed using simple comparison, it’s more complex phenomena (there are many additional variables that should be involved). It would be more appropriate to talk about potential or possible impact.

Specific comments:

Line 127:”The indoor air quality assessment was carried out in 30-minute series, with sampling every minute, both in the morning and in the afternoon” .  Does it mean that the 2 samples were performed on each day? It would be useful to add information (table) consisting of number of samples of each parameter with respect to each dwelling. I would suggest to include the Supplementary material containing summary of the measurements with respect to the dwellings

Line 148:” The statistical tests applied were the following: Pearson's Linear Correlation Coefficient, Pearson's Chi-square, Fisher's Exact Test and Ratio of Cross Products (Odds Ratio). “  It seems that not all of the tests were presented, please check

Figure 4. Please set 0 as starting value for concentration on Y scale (not a negative value)

Figure 6: Please add on the graph the protection threshold (included in the legend)

Figure 6 and Figure 10: The concentrations of outdoor PM10 and PM2.5 are presented, but it seems that for some dwellings PM2.5 is greater than PM10 (of the same dwelling, for example number 5, 7, …). It is obvious for maximum values from the graphs (hard to assess if it is the same for average). This is very strange and can’t be true (PM2.5 are already included in PM10). Please check it carefully and give some explanation.

Line 318: “…The indoor/outdoor ratio in the housing varied between 0.3 and 10…”  This statement is not consistent with the results in table 3. Please check it (also please check the CO2 values)

Line 87:”.. when the workers returned to their workplace at the HEI after ...” Please provide the full name for HEI when it appears for the first time in the main text

Typing errors

Please use PM10 – 10 in subscript

Author Response

First of all, we would like to express our deepest thanks to your very valuable and constructive comments to improve our manuscript.

Round 2

Reviewer 2 Report

Thank you for your responses to the comments raised in the previous version.